# Effect of Graphene on Modified Asphalt Microstructures Based on Atomic Force Microscopy

**DOI:** 10.3390/ma14133677

**Published:** 2021-07-01

**Authors:** Xian Li, Yanmin Wang, Yanling Wu, Huiru Wang, Qingliang Wang, Xingxing Zhu, Xiaocun Liu, Huadong Sun, Liang Fan

**Affiliations:** 1School of Traffic and Civil Engineering, Shandong Jiaotong University, Jinan 250300, China; lixian12132021@163.com (X.L.); 204145@sdjtu.edu.cn (Y.W.); whr4159@139.com (H.W.); QL51207@163.com (Q.W.); zhuxing5297@163.com (X.Z.); 204127@sdjtu.edu.cn (X.L.); 204113@sdjtu.edu.cn (H.S.); 2Shandong Transportation Institute, Jinan 250031, China; fanliang218@sina.com

**Keywords:** AFM, graphene, bee structures, phase transformation theory, anti-aging performance

## Abstract

Atomic force microscopy (AFM) was used to explore the effects of graphene modifier on the microstructure of asphalt. The morphologies of the before- and after-aged base asphalt and modified asphalt were performed and compared with analysis. The formation mechanism of asphaltic “bee structures” and the influence mechanism of graphene on asphalt were discussed from the classical theory of material science (phase transformation theory and diffusion theory). The results show that graphene facilitates the nucleation of “bee structures”, resulting in an increasing number and decreasing volume of “bee structures” in modified asphalt. Additionally, the anti-aging performance of the modified asphalt improved significantly because of graphene incorporation.

## 1. Introduction

Asphalt is a complicated mixture of various hydrocarbons and their nonmetallic (oxygen, sulfur, and nitrogen) derivatives. The components of asphalt include alkanes, cycloalkanes, arenes and sulfur-containing derivatives, polycyclic aromatic hydrocarbons, fused polycyclic arenes, and nonmetallic derivatives. Additionally, trace metal elements (vanadium, nickel, magnesium, iron, and calcium) are present in asphalt [1,2]. Although the components and microstructure of asphalt are crucial and affect its pavement performance, microscopic studies pertaining to asphalt are scarce owing to its complex chemical composition and insufficient research resources. In recent years, the relationship among bitumen components, microstructure, and pavement performance has garnered increasing attention. In particular, researchers are focusing on atomic force microscopy (AFM), which is used to investigate the microscopic structure and mechanism of asphalt.

In 1996, Loeber et al. first observed the microstructure of asphalt via AFM, and named the microscopic topography of asphalt as “bee structure.” By analyzing the formation process, it was preliminarily discovered that the “bee structure” was due to asphaltene [3]. Pauli et al. demonstrated that the microstructures are primarily due to the interaction between the crystallizing paraffin waxes and the remaining non-wax asphalt components [4]. Similarly, Jäger A. et al. reported that the “bee structure” was due to asphaltene [5]. De Moraes et al. speculated that the property of the “bee structure” is similar to that of microcrystalline wax [6]. Masson et al. investigated the microscopic topographies of 13 types of asphalt and discovered that the formation of “bee structures” was attributed to the content of vanadium and nickel in asphalt [7]. Meanwhile, it was reported that the addition of Trinidad Lake asphalt and SBS modifiers significantly affected the grain size and the distribution of “bee structures” [8,9]. Some researchers reported that the “bee structure” was due to the wrinkling of thin films measuring approximately 10 nm thick [2,10]. Ji et al. used AFM to quantitatively evaluate the microstructure of asphalt via roughness theory [11]. Hung investigated the evolution of the microstructure of asphalt after water exposure [12]. Although achievements as evidenced via AFM have been realized in the microscopic application of asphalt, the formation conditions, mechanism, evolution law, and influencing factors of asphaltic “bee structures” remain inconclusive. Therefore, we try to explore the effect of additives on the microstructure of asphalt and expect to achieve something meaningful.

“Small size effect” nanomaterials naturally have a boundary effect and high specific surface area, which can directly or indirectly lead to a special surface effect. In recent years, nanomaterials used as asphalt modifications have increasingly been studied to improve the mechanical and physical properties of bitumen. Some nanomaterials such as titanium dioxide (TiO_2_) [13], zinc oxide (ZnO) [14], graphene oxide (GO) [15], silicon dioxide (SiO_2_) [16], and montmorillonite [17] have been selected as modifiers to improve the road performance and extend the durability of asphalt. Some very significant results have been achieved. For example, Cheraghian et al. studied the ultraviolet aging resistance of fumed silica nanoparticles modified bitumen and the results show that the anti-ultraviolet aging property of modified asphalt is improved with the increase of nano silica content [18]. Zhang et al. found that the reduction of ZnO particle size can improve the performance of asphalt binder and asphalt mixture through analyzing the influence of ZnO particle size on asphalt’s physical properties [19].

In our study, graphene was selected as an asphalt modifier. This is mainly because graphene is a novel quasi-two-dimensional (2D) carbon-based nanomaterial with carbon atoms arranged in a honeycomb lattice, and has been investigated as an asphalt modifier owing to its compatibility with asphalt. For example, Wu et al. discovered that incorporating a small amount of graphene oxide (GO) improved the anti-aging property of asphalt [20,21,22]. Shi et al. investigated the properties and modification mechanism of GO-modified asphalt and discovered that a trace amount of GO restricted the movement of asphalt molecules and improved their high-temperature performance [23,24,25]. Moreno–Navarro et al. discovered that the presence of graphene resulted in a more significant elastic response and reduced the thermal sensitivity of the asphalt binder [26].

Therefore, in this study, the topographies of base A-70 asphalt and graphene-modified asphalt were investigated via AFM. A more comprehensive analysis and discussion would clarify the formation reasons and the effects of graphene on asphaltic “bee structures”. Basic material rules, phase transformation theory, and diffusion theory were introduced to interpret the mechanism of graphene on asphalt’s microscopic “bee structures.”

## 2. Experimental

In this study, A-70 asphalt provided by Shandong Hi-speed Construction Materials, Jinan, China, was used as the base asphalt. Graphene-modified asphalt was home-made and the dosage of graphene was 1% in weight, which has previously been proven to be the optimal graphene content [27].

### 2.1. Preparation of AFM Specimens

The short- and long-term aging of asphalt were simulated using a rotating film oven heating apparatus (James Cox & Sons CS325B, Colfax, CA, USA) and a pressure aging apparatus (Prentex PR9300, Sunnyvale, TX, USA), respectively. Unaged, short-term aged, and long-term aged asphalt were prepared separately. AFM specimens were prepared via the heating molding method as follows: First, molten asphalt was deposited onto a glass slide (10 mm × 10 mm × 1 mm) using a glass rod. Subsequently, the drop was placed in an oven and tilted 30° above the horizontal. It was heated at 150 °C for 15 min, and a thin layer of asphalt coated the surface of the slide by gravity. Finally, the glass slide coated with asphalt was cooled to room temperature in the air. The asphalt film prepared via this method measured several microns in thickness, which is similar to the actual thickness of asphalt films on roads [28].

### 2.2. Atomic Force Microscope Test

AFM (Bruker Multimode 8, Santa Barbara, CA, USA)was performed to characterize the microstructures of the base and graphene-modified asphalt before and after aging. A schematic diagram of the AFM performed is shown in Figure 1. The four core components used in the AFM were a probe tip, cantilever, laser, and position-sensitive photodiode (PSPD). During scanning, the height variation of the microsurface of the sample will produce a slight force, i.e., an attraction or repulsive force between the probe tip and sample surface. In this case, the cantilever deviates based on “Hooke’s Law,” resulting in a change in the reflection signal of the laser illuminated on the back of the cantilever. The change in the laser reflection signals can be sensed by the PSPD, and the surface topography of the sample can be obtained after treatment [29]. The topography and phase-contrast images were received in the intermittent contact mode, where the elastic constant of the probe was 0.4 N/m, scanning rate was 1.5 Hz, scanning area was 10 μm × 10 μm, and the resolution was 10 nm.

## 3. Results and Discussion

### 3.1. Effect of Graphene on Fundamental Properties of Asphalt

Figure 2a,b show the basic properties of the base and graphene-modified (the dosage of graphene was 1.0% in weight) asphalt before and after aging, respectively. As shown in Figure 2, both the base and modified asphalt show a trend of decreasing penetration, increasing softening point, increasing viscosity, and decreasing ductility after short-term rolling thin film oven test (RTFOT) aging. The increasing softening point and viscosity indicate that the asphalt becomes hard and brittle after aging. Meanwhile, the anti-aging property of asphalt could also be evaluated by comparing the change rates of the indexes before and after aging. It can be concluded that the changing rate of every index of graphene-modified asphalt was lower than that of the base asphalt. This suggests that the anti-aging performance of the graphene-modified asphalt was superior to that of the base asphalt. Additionally, the high-temperature performance of the modified asphalt improved because of the incorporation of graphene.

### 3.2. Effect of Graphene on “Bee Structure”

Figure 3a,b show the 2D topographies of the base and graphene-modified asphalt measured via AFM, respectively. Three phases (Catana, peri, and para phases) were observed from the images shown in Figure 3. The “bee structure” was composed of interlacing light and dark lines, which represent convex and concave structures, respectively [7,30]. Furthermore, Figure 3a,b show many “bee structures” in the graphene-modified asphalt within the same investigation area. The height of the light, bright peak increased significantly, and the depth of the dark valley decreased, indicating a decrease in the size of the “bee structure” in the graphene-modified asphalt. In other words, in the base asphalt, the “bee structures” are larger but fewer in quantity.

AFM provides topographic information in the form of height lines (or profiles), e.g., the distance between the higher and lower parts of the “bee structure.” Figure 3c,d show topographic profiles of the selected representative “bee structures” labeled in Figure 3a,b, respectively. As shown, the average length of the “bee structure” in the base asphalt was 3–4 μm, whereas it is reduced to 2–3 μm in the graphene-modified asphalt. Additionally, the average altitude of the “bee structure” in base asphalt was approximately 44.2 nm, whereas it was reduced to approximately 35.4 nm in the graphene-modified asphalt. This reduction shows that the volume of the “bee structure” in asphalt decreased after modification by graphene. By analyzing Figure 3, it can be speculated that the decline in the “bee structure” volume in the graphene-modified asphalt is primarily related to the factors contributing to the formation of the “bee structures.”

### 3.3. Effect of Graphene on Microstructure during Asphalt Aging

Figure 4 shows the three-dimensional (3D) AFM images of the base and graphene-modified asphalt before and after aging. As shown, the surface of the graphene-modified asphalt was smooth, whereas that of the base asphalt was rough. After rolling-thin-film-oven (RTFOT) aging, the variation in the base asphalt surface was more evident than that of the graphene-modified asphalt. Specifically, the peak area increased significantly, and the quantity and size of the “bee structures” increased as well. However, the AFM images of the graphene-modified asphalt before and after aging were similar. The number of “bee structures” in the base and graphene-modified asphalt increased after pressure-aging vessel (PAV) tests. This occurred because the asphalt viscosity increased and the number of large aromatic hydrocarbon molecules increased after aging [31].

Quantitative analysis data including roughness, number peaks found, and minimum and maximum peak depth can be extracted from Figure 4 through “AFM software—Nanoscope Analysis” and the list in Table 1.

Firstly, the variation in the surface morphology can be quantitatively analyzed by the difference between the microscopic phase states of asphalt based on the roughness index (root-mean-square roughness (*R*_q_), average roughness, and maximum height roughness) [32]. In this study, the relationship between the microscopic phase states and *R*_q_ was investigated. *R*_q_ was calculated as follows:(1)$$R_{q}=\sqrt{\frac{\iint {\left[h\left(x,y\right)-h_{0}\right]}^{2}dA}{\iint dA}}$$
(2)$$h_{0}=\frac{\iint h\left(x,y\right)dA}{\iint dA}$$
where *A* is the scanning area, which measured 10 μm × 10 μm in this study; *h* (*x*, *y*) is the height function of the morphology (nm); and *h*_0_ is the reference height (nm). The *R*_q_ of each sample listed in Table 1 was obtained using “AFM software—Nanoscope Analysis—Roughness,” and was also presented in forms of graphs, as shown in Figure 5.

As shown in Figure 5, the *R*_q_ of the graphene-modified asphalt was smaller than that of asphalt, indicating that the addition of graphene reduced the roughness of the asphalt surface. The surface roughness is related to the self-healing ability and adhesion performance of asphalt [33]. Briefly, the higher the roughness of the asphalt binder, the better the self-healing ability and adhesion performance of the asphalt. The roughness increased after short-term aging, whether in the base asphalt or graphene-modified asphalt [34,35]. However, the roughness decreased after long-term aging. The different effects of graphene on the AFM morphology and roughness of asphalt after short- and long-term aging were primarily attributed to the decrease in the light components (saturated hydrocarbon and aromatic hydrocarbon) and, correspondingly, the increase in asphaltene and resin contents [36]. In this case, more asphaltenes were attached to the wax chips during the formation of the “bee structure,” resulting in an increase in the “bee structure” volume and the AFM roughness of both the base and graphene-modified asphalt after short-term aging.

It was observed that the *R*_q_ of the two types of asphalts increased after RTFOT aging but decreased after PAV aging. This may be because the aging degree of RTFOT was lower, a smaller number of light components were volatilized, and the heavy components were exposed, thereby causing an increase in the surface roughness. Meanwhile, the aging degree of PAV was higher, thereby resulting in a larger number of lightweight elements to be evaporated; consequently, the asphalt transitioned from a multiphase structure to a single-phase structure. The simplification of the phase state reduced the surface roughness. Although the *R*_q_ of the two types of asphalts increased after RTFOT aging, the increment in the graphene-modified asphalt was smaller than that of the base asphalt. After PAV aging, the *R*_q_ of the two types of asphalts decreased, but the reduction in the graphene-modified asphalt was less than that of the base asphalt. Therefore, it can be inferred from *R*_q_ that graphene addition enhanced the anti-aging ability of the base asphalt.

Secondly, the variation in the surface morphology can be quantitatively analyzed by the number of peaks found and peak depth. A group of peaks associate to form a bee structure. The number of bee structures has been previously discussed and here the number of peaks found have not been talked about. A frequency histogram of peak depth distributions is constructed, as presented in Figure 6. As shown, the peak depth distribution histogram of the graphene-modified asphalt under different aging conditions exhibited a smaller peak line than that of the base asphalt, indicating that the number of peaks with greater depth in the graphene-modified asphalt was fewer and the distribution occupied a smaller proportion of the area. The peak heights of the unaged, short-term aged, and long-term aged base asphalt were concentrated in the vicinity of 70, 83, and 62 nm, respectively. The peak heights of the unaged, short-term aged, and long-term aged graphene-modified asphalt were concentrated in the vicinity of 47, 60, and 57 nm, respectively. The addition of graphene might have affected the accumulation of structural components in the “bee structures,” thereby reducing the peak height. The height of the after-aged “bee structure” decreased owing to the change in the asphalt components caused by asphalt aging, the decrease in non-polar components (saturated and aromatic), and the increase in polar features (colloids and asphaltenes).

### 3.4. Discussion of Asphalt Microstructure Based on Liquid-Solid Phase Transformation Theory

In our opinion, the formation of “bee structures” and the effect of graphene on the “bee structures” can be elucidated using the basic theory of “liquid–solid phase transition”. According to thermodynamic equilibrium theory, a phase transition can occur to form a new phase when the material is cooled to the phase transition temperature. Asphalt, an extremely complex organic mixture, appears in the molten state at high temperatures. During cooling, some components of molten asphalt can undergo liquid–solid phase transition, which results in phase separation in asphalt. The “bee structure” is the result of phase separation in asphalt. According to Gibbs, the phase change process can be categorized into two, i.e., nucleation-growth phase change and continuous phase change. In our opinion, the “bee structure” formation of asphalt belongs to nucleation-growth phase change. Moreover, the phase change process can be categorized into diffusive phase change or non-diffusive phase change based on the lattice migration characteristics. In our opinion, the “bee structure” formation of asphalt is a crystallization process that belongs to diffusive phase change. Therefore, the “bee structure” formation and the effect of graphene on the “bee structure” can be briefly analyzed as follows.

#### 3.4.1. Analysis of “Bee Structure” Formation

The formation of the “bee structure” has been debated, and in our opinion, the “bee structures” of asphalt are due to wax crystallization or associations involving wax and other concretes (such as asphaltenes and modifiers). During the freezing of asphalt, crude and residue oil—hydrocarbon mixtures with high melting points capable of precipitation via crystallization—are collectively known as wax. Wax is vital to the formation of the “bee structure.” The appearance of needle-flake crystals after cooling signifies the presence of microcrystalline wax in asphalt [37].

Wax crystallization involves the transition from a nuclear embryo to a crystal nucleus, followed by a crystal. During cooling, the alkane molecules that were distributed randomly in molten asphalt change from a high free energy state (liquid state) to a low free-energy state (crystalline state), in which the alkane molecules within the short-range are arranged orderly to form nucleus embryos of the “nuclear embryo”, which facilitates the further formation of a stable crystal nucleus. A nuclear embryo is a prerequisite for the construction of a nucleus. However, the nuclear embryo will disintegrate if the temperature increases, whereas it will form a stable nucleus that will enlarge and develop crystals if the melt continues to be cooled. The crystallization process comprises nucleation formation and grain growth, both of which require the appropriate supercooling degree. As the temperature decreases, these molecules will undergo a continuous connect–fracture and fracture–connect process to form ordered lattice points until a critical size is attained (a new stable state), i.e., a crystal nucleus [38]. Finally, other surrounding molecules will always cover the crystal lattice points and gradually form a thin slice structure that approaches the crystal nucleus and causes the crystal nucleus to develop into a needle-shaped crystal.

The growth of crystal occurs in the region where the polymerization energy between the crystal and free paraffin is the greatest, resulting in the fastest growth of the sheet structure located on the side of the crystal nucleus. In the other components of the microcrystalline wax and asphalt, asphaltene molecules serve as the core for accumulating crystal clusters and then further develop them into “bee structures.” When the asphalt system is cooled to below the crystallization temperature, the oil in the asphalt on both sides of the peak sheet will ascend along the peak, which can be considered a capillary phenomenon in molten asphalt [39]. Figure 7 shows a schematic diagram of the precipitation of the “bee structures.” In the molten state, asphaltic components (saturates, aromatics, resins, and asphaltenes) are mixed to a homogeneous state. Subsequently, through modification, the graphene modifier is uniformly dispersed in the molten asphalt to form a homogeneous system. During cooling, graphene and asphaltenes will become nucleation sites, and the wax will crystallize easily, thereby resulting in “bee structures.”

#### 3.4.2. Effect of Graphene on the Bee Structures of Asphalt

Previous conclusions indicated that the modified graphene “bee structures” appeared in a greater quantity and were smaller than those of the base asphalt. The formation of a crystal nucleus is the first step in crystallization, and the nucleation process can be categorized into inhomogeneous and homogeneous nucleation based on crystal nucleation theory. Homogeneous nucleation refers to the same probability of nucleus generation in undercooled melts. Meanwhile, inhomogeneous nucleation refers to a formation process facilitated by various catalytic positions such as the surface, interface, cracks, and walls.

The body of the stable graphene modifier will become the catalytic site for nucleus formation in modified asphalt, and this is classified under inhomogeneous nucleation. The incorporation of graphene provides numerous nucleation sites, and interfaces provide regular (spherical) templates on which wax molecules can be deposited [40]. The barrier of inhomogeneous nucleation (∆*G_K_**) is less than that of homogeneous nucleation (∆*G_K_*), and the relationship exists in the asphalt, as shown in Equation (3), where *θ* represents the contact angle between a cap-shaped nucleus and a flat substrate, as depicted in the classical nucleation theory. Figure 8 shows the cap-shaped model of inhomogeneous nucleation. *Cos θ* can be calculated from Young’s equation (Equation (4)). In Equation 4, *γ_nl_*_,_
*γ_sl_*_,_ and *γ_sn_* refer to the interfacial free energies between the nucleus and liquid, substrate and liquid, and substrate and nucleus, respectively. f(θ) can be obtained from Equation (5) of the geometric relationship of the cap model, and its value is less than or equal to 1.
(3)$$\Delta G_{K}^{*}=\Delta G_{K}f(\theta )$$
(4)$$\cos\theta =\frac{\gamma_{sl}-\gamma_{sn}}{\gamma_{nl}}$$
(5)$$f(\theta )=\frac{(2+\cos\theta ){\left(1-\cos\theta \right)}^{2}}{4}\leq 1$$

When the crystal nucleus is formed on the nucleating agent, the nucleation barrier decreases with the contact angle (*θ*), and the inhomogeneous nucleation barrier is lower than the homogeneous nucleation barrier, which facilitates crystallization. In the base asphalt, asphaltene can serve as a nucleating agent. By contrast, in the graphene-modified asphalt, uniformly dispersed graphene in asphalt will share its role as a nucleating agent with asphaltene. Although wax can be detached via inhomogeneous nucleation in both the base and modified asphalt, the number of nucleating agent particles in both cases will differ. In our opinion, the graphene modifier can serve as an additional dispersed nucleation center that facilitates the formation of a large number of smaller wax crystals [41]. Therefore, the number of “bee structures” in the graphene-modified asphalt will be higher than that of the base asphalt. Additionally, graphene can result in a lower “bee structure” volume owing primarily to the formation of a relatively compact gel network in the modified asphalt. In our opinion, the formation of the “bee structures” is categorized under diffusive phase transition, and the increased viscosity of modified asphalt hinders the diffusion and transfer of wax molecules.

The growth stage of the “bee structure” can be explained based on diffusion theory. A detailed explanation has been presented in our previous study [27]. In asphalt, a lower viscosity results in less intermolecular interactions, whereas a smaller resistant force toward migration results in a higher molecular migration rate, which facilitates the migration of asphalt components. The viscosity of the base asphalt is less than that of the graphene-modified asphalt. The wax components can migrate quickly in the base asphalt, thereby facilitating the development of the “bee structure.” Meanwhile, the number of nucleation sites in the base asphalt is less than that of the graphene-modified asphalt. Therefore, the asphaltic “bee structures” of the base asphalt enlarges, and their distributions are scattered. Meanwhile, the viscosity of the modified asphalt is high, resulting in the low migration speed of the wax molecules. Furthermore, graphene as an asphaltene can serve as a nucleation site and impede particle migration. The factors above can result in the abundant quantity and smaller size of “bee structures” in modified asphalt.

## 4. Conclusions

The micromorphology of unaged and aged base asphalt and graphene-modified asphalt were investigated via AFM. The micrograph variations were compared and analyzed. The formation mechanism of asphaltic “bee structures” and the effect of graphene on the “bee structures” were discussed. The main conclusions obtained are as follows:

(1) Graphene can serve as additional dispersed nucleation centers that facilitate the formation of numerous smaller “bee structures”;

(2) The micrograph variation of the graphene-modified asphalt after aging was smaller than that of the base asphalt. Moreover, the *R*_q_ of the graphene-modified asphalt with different aging degrees was lower than that of the base asphalt, indicating that graphene addition improved anti-aging performance;

(3) The formation of “bee structures” in asphalt can be explained as follows: During the cooling of asphalt, alkanes were arranged to form a nucleus embryo, which then underwent nucleation and growth. Additionally, graphene affected the nucleation and growth processes;

(4) Basic material rules, phase transformation theory, and diffusion theory were introduced to analyze the growth morphology of the “bee structures”.

## Figures and Tables

**Figure 1 materials-14-03677-f001:**
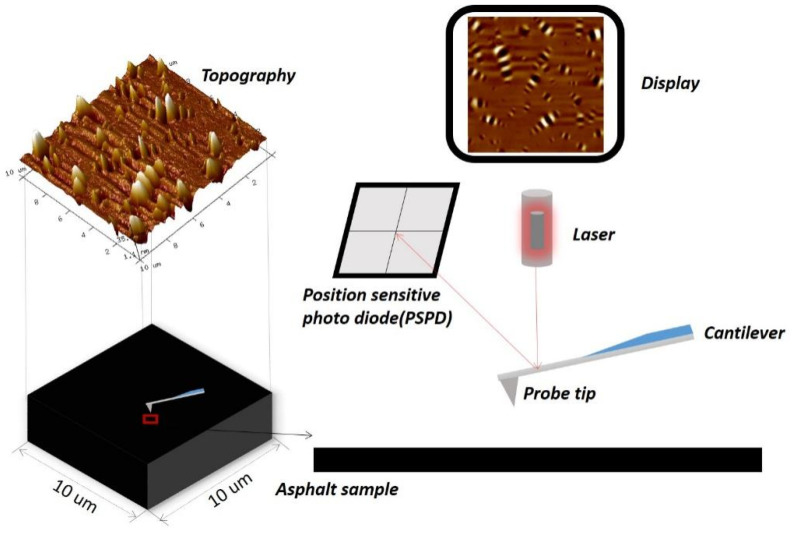
Schematic diagram of AFM.

**Figure 2 materials-14-03677-f002:**
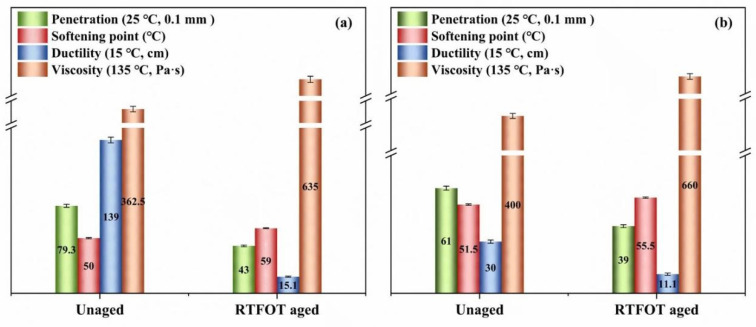
Basic properties of base asphalt (**a**) and graphene-modified asphalt (**b**) before and after aging.

**Figure 3 materials-14-03677-f003:**
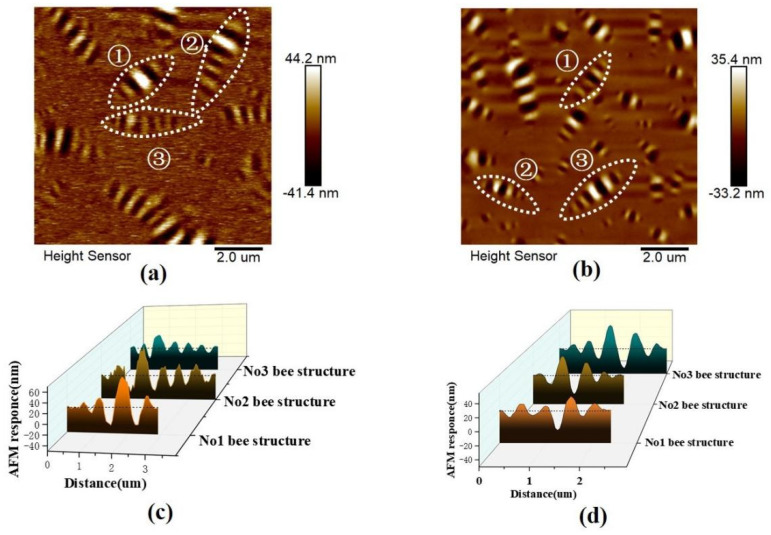
Topographies of asphalt investigated via AFM. (**a**) 2D image of base asphalt; (**b**) 2D image of graphene-modified asphalt; (**c**) topographic profiles of “bee structures” labeled in (**a**); (**d**) topographic profiles of “bee structures” labeled in (**b**).

**Figure 4 materials-14-03677-f004:**
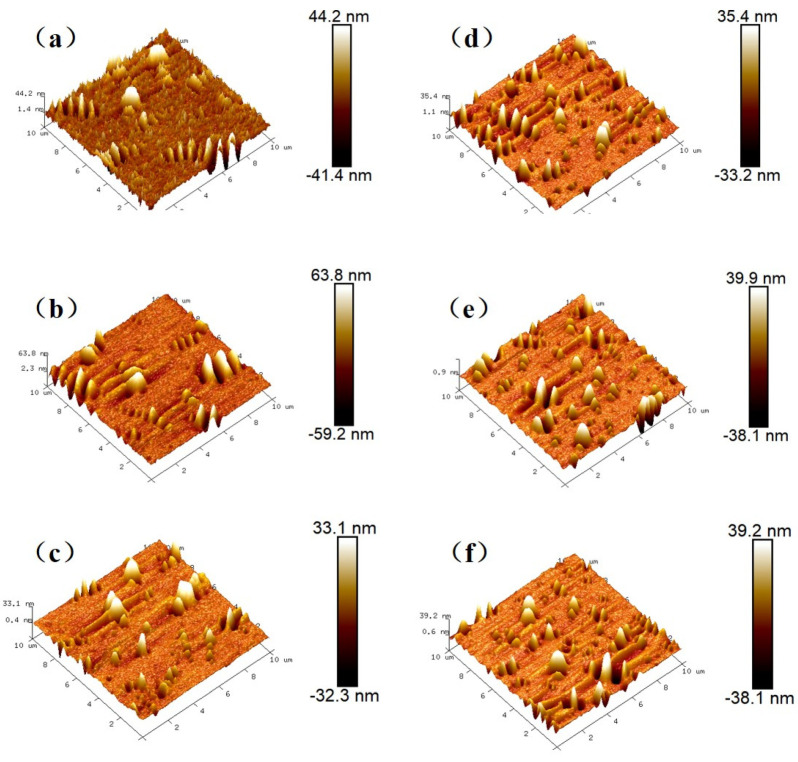
(**a**) 3D AFM images of base asphalt; (**b**) base asphalt after RTFOT aging; (**c**) base asphalt after PAV aging; (**d**) graphene-modified asphalt; (**e**) graphene-modified asphalt after RTFOT aging; (**f**) graphene-modified asphalt after PAV aging.

**Figure 5 materials-14-03677-f005:**
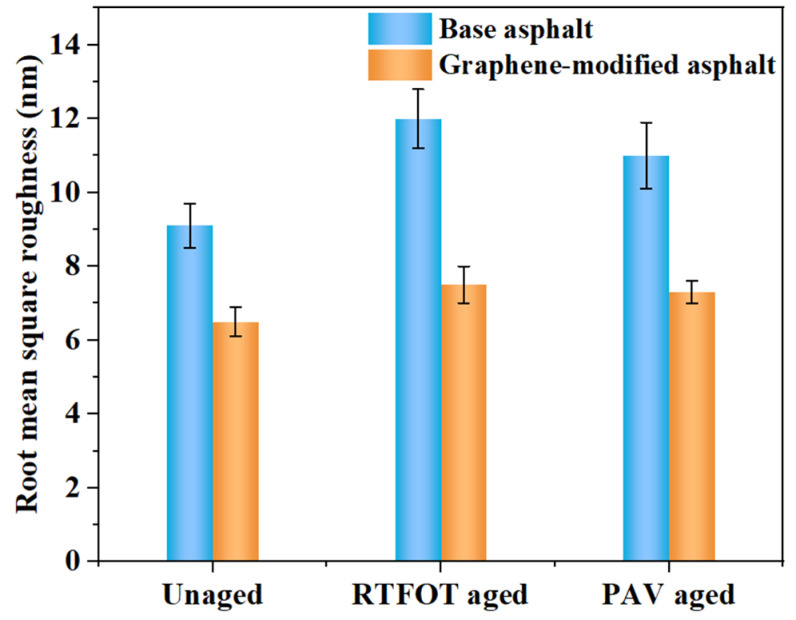
Microscopic surface roughness index of asphalt.

**Figure 6 materials-14-03677-f006:**
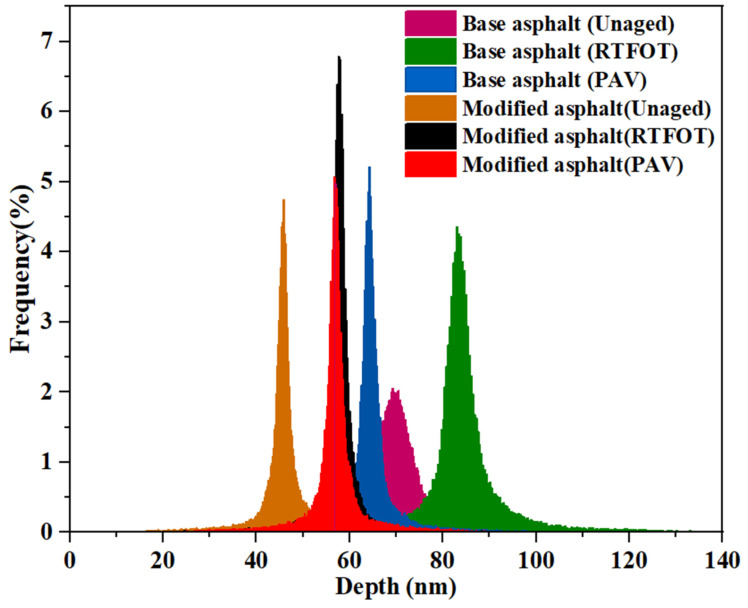
Height distribution of asphalt micromorphology.

**Figure 7 materials-14-03677-f007:**
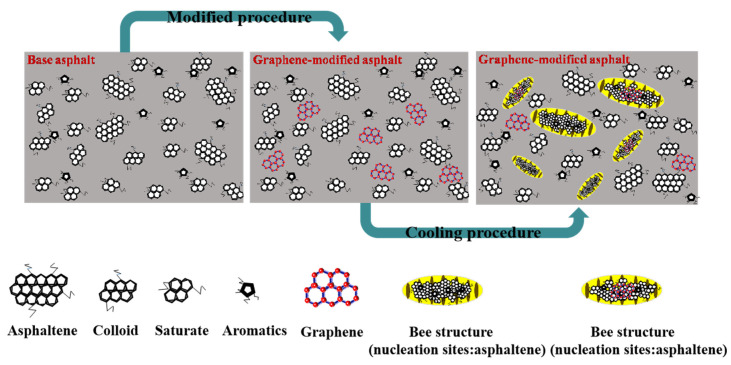
Schematic diagram of precipitation process of “bee structures”.

**Figure 8 materials-14-03677-f008:**
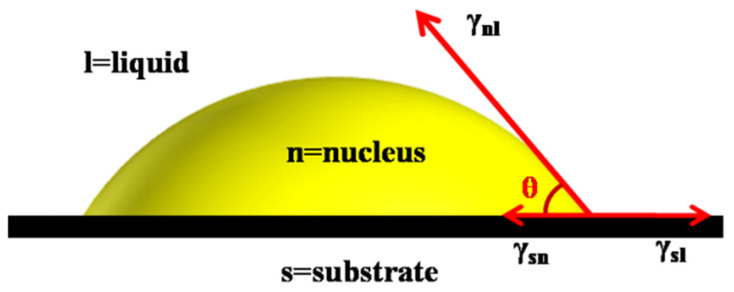
Cap-shaped model of inhomogeneous nucleation.

**Table 1 materials-14-03677-t001:** Basic data parameters extracted from the 3D AFM topographies.

Indicators			Base Asphalt	Graphene-Modified Asphalt
Aging states	Unaged	Roughness (nm)	9.1	6.5
		Number Peaks Found (number)	141.0	137.0
		Minimum Peak Depth (nm)	70.5	48.4
		Maximum peak Depth (nm)	69.2	45.3
	RTFOT	Roughness (*R*_q_)	12.0	7.5
		Number Peaks Found (number)	133.0	140.0
		Minimum Peak Depth (nm)	77.9	67.9
		Maximum Peak Depth (nm)	83.1	63.9
	PAV	Roughness (*R*_q_)	11.0	7.3
		Number Peaks Found (number)	130.0	137.0
		Minimum Peak Depth (nm)	53.3	68.3
		Maximum Peak Depth (nm)	57.7	64.5

## Data Availability

The data is available on the request to the corresponding author.

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
