# Peer review of "Effect of Graphene on Modified Asphalt Microstructures Based on Atomic Force Microscopy"

_materials, 2021, doi:10.3390/ma14133677_

Round 1

Reviewer 1 Report

  • The abstract needs to be improved. You need some short introduction, objectives, and major conclusions.
  • Line 22: "extremely complicated" is a very abstract statement for a scientific article. 
  • The literature review is very weak. You need more information. 
  • Your paper has no objectives.
  • Figure 2: I am not sure what this figure is about.
  • Figure 4: You need some quantitative analysis
  • Figure 5: too big

Overall, it is an interesting paper, but it needs further improvements. 

Reviewer 2 Report

The paper “Effect of graphene on modified asphalt microstructures based on atomic force microscopy” investigated topographies of base A-70 asphalt and graphene-modified asphalt by AFM. The manuscript is well organized; however, to improve the quality, the following recommendations can be incorporated.

1.The authors should review the other more new investigation on their study way in the introduction part and finally note the novelty of the article. The introduction part needs to develop.

  1. Authors can cite the following work in the introduction which is closely related to their work and recently reported:

- "The utilization of graphene oxide in traditional construction materials: Asphalt." Materials 10, no. 1 (2017): 48./"Effect of Fumed Silica Nanoparticles on Ultraviolet Aging Resistance of Bitumen." Nanomaterials 11, no. 2 (2021): 454.

3- Explain the results obtained from the Effect of graphene on “bee structure” more thoroughly.

4- Technical terms are misused through the manuscript and the writing needs a revision.

5- What is the purpose of the authors in using the concept of wettability (Cap-shaped model) needs further explanation and testing.

6- Most of the references are old, new articles should be used as much as possible.

7- The authors have to say exactly what is the difference between the base sample and the aged sample. There is no difference in the AFM figures.

Round 2

Reviewer 1 Report

Thank you for addressing my comments.

Author Response

Thanks again for the reviewer's comments.

Reviewer 2 Report

The remarks I have posted were generally meant or suggested to be included in the article, not to be explained to the reviewer. Some of them were just addressed by the Authors in form of answers to the reviewer, but in my opinion, they should be at least partly included in the article. I really would like to encourage the Authors to implement much more details in your manuscript for better understanding not only by road engineers. Journal of Materials is not the dedicated journal for the road engineering society only. The potential readers might not achieve a good understanding of the information provided in the article if it is not well explained. My remarks are meant to help you improve the presentation of your valuable research. 
The answer to points 1, 2 and, 5 of the previous round of reviewing is not seen in the manuscript.
